# Application of Statistical K-Means Algorithm for University Academic Evaluation

**DOI:** 10.3390/e24071004

**Published:** 2022-07-20

**Authors:** Daohua Yu, Xin Zhou, Yu Pan, Zhendong Niu, Huafei Sun

**Affiliations:** 1School of Computer Science and Technology, Beijing Institute of Technology, Beijing 100081, China; yudaohua@bit.edu.cn; 2School of Mathematics and Statistics, Beijing Institute of Technology, Beijing 100081, China; 3120211473@bit.edu.cn (X.Z.); p4nyu@foxmail.com (Y.P.); huafeisun@bit.edu.cn (H.S.); 3School of Computing and Information, University of Pittsburgh, Pittsburgh, PA 15260, USA

**Keywords:** statistical K-means, academic evaluation, statistical manifold, clustering

## Abstract

With the globalization of higher education, academic evaluation is increasingly valued by the scientific and educational circles. Although the number of published papers of academic evaluation methods is increasing, previous research mainly focused on the method of assigning different weights for various indicators, which can be subjective and limited. This paper investigates the evaluation of academic performance by using the statistical K-means (SKM) algorithm to produce clusters. The core idea is mapping the evaluation data from Euclidean space to Riemannian space in which the geometric structure can be used to obtain accurate clustering results. The method can adapt to different indicators and make full use of big data. By using the K-means algorithm based on statistical manifolds, the academic evaluation results of universities can be obtained. Furthermore, through simulation experiments on the top 20 universities of China with the traditional K-means, GMM and SKM algorithms, respectively, we analyze the advantages and disadvantages of different methods. We also test the three algorithms on a UCI ML dataset. The simulation results show the advantages of the SKM algorithm.

## 1. Introduction

University academic evaluation involves using different indicators and methods to measure the academic level of universities. It has great motivating, guiding and restricting effects on the development of universities, thus gaining more and more attention nowadays [1,2,3,4]. In [5], the authors proposed a statistical method of constructing an evaluation system for the transformation of scientific and technological achievements by using Principal Component Analysis (PCA) and the comprehensive indicator method. In [6], the authors used Decision-making Trial and Evaluation Laboratory (DEMATEL) and the entropy-weighting method to give assessments on the research innovation ability of universities in a subjective and objective way. In [7], the authors used the Analytic Hierarchy Process (AHP) method to design the evaluation indicators and give the corresponding weights. However, these works are all based on the specific design of weighted indicators, which cannot avoid the interference of the subjective thoughts of the evaluators and highly depend on the type of universities. In addition, with the development of big data, more and more statistic data are generated yet not properly used, as it is hard to attribute weights for so many indicators. In this paper, we introduce the statistical K-means algorithm to give the academic evaluation results of universities. The idea is mapping the evaluation data together with the clustering problem from Euclidean space to Riemannian space. Specifically, the local statistics are used as parameters to determine a special parameter distribution, which projects all data points into parameter space to obtain a parameter point cloud. This idea has been well applied in many research fields. In [8], the authors take a step forward in image and video coding by extending the well-known Vector of Locally Aggregated Descriptors (VLAD) onto an extensive space of curved Riemannian manifolds. In [9], the authors propose a method which allows us to fuse information from feature representations from both Euclidean and Riemannian spaces by mapping data in a Reproducing Kernel Hilbert Space (RKHS). This method achieves state-of-the-art performance on the problem of pose-based gait recognition. These findings suggest that this idea has great value and significance in the information field. In this paper, our main contributions can be summarized as two points. Firstly, we use statistical manifolds theory to extract features from the origin point cloud, which is capable of processing the high-dimensional data and proves to be a great substitution of the traditional method PCA. Secondly, we use clustering methods to give an evaluation on the academic level of Chinese universities instead of scoring or rating. With the change of the cluster numbers, the underlying relationships of universities in terms of subject development can be found, and the academic level can be assessed by the clustering results subjectively. These two points also provide new research ideas for related problems.

The paper is organized as follows. In Section 2, we introduce some basic knowledge about multivariate normal distribution manifold, difference functions and Gaussian mixture models. In Section 3, we introduce the local statistical methods and statistical K-means (SKM) algorithm. In Section 4, we describe the work of data pre-processing, including the data source and data pre-processing strategies, and we introduce the criteria for assessing the clustering algorithms. In Section 5, we conducted the simulation experiments with the traditional K-means, GMM and SKM algorithm for the top 20 universities of China and analyze their advantages and disadvantages, respectively. A UCI ML dataset is also tested to quantitatively measure the algorithms.

## 2. Preliminary

### 2.1. Multivariate Normal Distribution Manifold

Information geometry is used to solve some nonlinear and stochastic problems in the information field, because compared with the treatment in the Euclidean space, the one of Riemannian manifold can often achieve precise results. The statistical manifold is a set of all probability density functions with some regular conditions. In addition, by introducing the Fisher information matrix as a Riemannian metric, the statistical manifold becomes a Riemannian manifold. It is well known that the Kullback–Leibler divergence is a suitable difference function measuring the difference of two points on the statistical manifold, even though it is not a real distance function [10,11]. The manifold of a family of multivariate normal distributions is an important statistical manifold and is widely applied to the researches of signal processing, image processing, neural networks and so on. The K-means algorithm on statistical manifolds introduced in this paper is to transform the data point cloud in Euclidean space into the parameter point cloud on the statistical manifold of a family of multivariate normal distributions, and then, it applies cluster analysis to the parameter point cloud.

**Definition** **1.**
*We call a set*

$$S=\left\{p\left(x;\theta \right)|\theta \in \Theta \subset R^{n}\right\}$$

*an n-dimensional statistical manifold, where p(x;θ) is the probability density of functions, with some regular conditions.*


Since each *n* multivariate normal distribution density function can be determined by an *n*-dimensional vector (mean) and an *n*-order symmetric positive definite matrix (covariance matrix), the manifold that consists of the family of normal distributions is closely related to manifold of the symmetric positive definite matrices [12].

**Definition** **2.**
*The manifold of symmetric positive definite matrices SPD(n) is defined as*

$$\begin{array}{c} \mathrm{SPD}\left(n\right)=\left\{P\in M\left(n\right)|P^{T}=P,\mathrm{and}\;x^{T}Px>0,\forall x\in R^{n}-\left\{0\right\}\right\}, \end{array}$$

*where M(n) is the set of n-order matrices and $P^{T}$ denotes the transpose of the matrix P. The smooth structure on SPD(n) is induced as the submanifold of the general linear group GL(n,R), which is a set of all non-singular matrices.*


**Definition** **3.**
*The multivariate normal distribution manifold consists of the probability density functions of all n multivariate normal distributions, which is defined as*

$$\begin{array}{c} N_{n}=\left\{f|f\left(\mu ,\Sigma \right)=\frac{1}{\sqrt{{\left(2\pi \right)}^{n}\det\left(\Sigma \right)}}\exp\left\{-\frac{{\left(x-\mu \right)}^{T}\Sigma^{-1}\left(x-\mu \right)}{2}\right\}\right\}, \end{array}$$

*where $\mu \in R^{n}$ and Σ∈SPD(n) are the mean and the covariance matrix of the distributions, respectively, and (μ,Σ) is called the parameter coordinate of $N_{n}$.*

*It is worth noting that $N_{n}$ is topologically homeomorphic in the product space $R^{n}\times \mathrm{SPD}\left(n\right)$.*


### 2.2. Difference Functions on Multivariate Normal Distribution Manifold

In this paper, we need to consider the difference between the probability density functions of different multivariate normal distributions. We select the Wasserstein distance as the difference function. At the same time, we also use Kullback–Leibler divergence, which is a difference function commonly used in classical information theory. We will introduce these difference functions respectively below [13,14,15].

#### 2.2.1. Wasserstein Distance

The Wasserstein distance of the probability measure on $R^{n}$ describes the energy required to transfer between the two distributions.

In particular, for the multivariate normal distribution, the literature [13] gives a specific expression.

**Proposition** **1.**
*The Wasserstein distance between $P_{1},P_{2}\in N_{n}$ is*

(1)
$$D_{W}^{2}\left(P_{1},P_{2}\right)={∥\mu_{1}-\mu_{2}∥}^{2}+\mathrm{tr}\left(\Sigma_{1}+\Sigma_{2}-2{\left(\Sigma_{1}\Sigma_{2}\right)}^{\frac{1}{2}}\right),$$

*where $\left(\mu_{1},\Sigma_{1}\right)$ and $\left(\mu_{2},\Sigma_{2}\right)$ correspond to the distribution of $P_{1}$ and $P_{2}$, respectively.*


Unfortunately, there is not a simply explicit expression of the geometric mean of the Wasserstein distance; hence, this paper temporarily replaces the geometric mean with the arithmetic mean in the simulation experiments.

#### 2.2.2. Kullback–Leibler Divergence

Kullback–Leibler (KL) divergence is a non-negative function which measures the difference between any two probability density functions. It is worth noting that KL divergence is not a distance function, since it does not satisfy the symmetry and triangle inequality. In the following, we give its definition and the expression of its geometric mean.

**Definition** **4.**
*Let $P_{1}$, $P_{2}$ be two probability density functions. KL divergence is defined as*

(2)
$$D_{KL}\left(P_{1}\left|\right|P_{2}\right)=E_{P_{1}}\left[\log\frac{P_{1}}{P_{2}}\right],$$

*and it can be shown that $D_{KL}\left(P_{1}\left|\right|P_{2}\right)\geq 0$; the equality holds if and only if $P_{1}=P_{2}$.*


In particular, for any $P_{1},P_{2}\in N_{n}$ with the parameters $\left(\mu_{1},\Sigma_{1}\right)$ and $\left(\mu_{2},\Sigma_{2}\right)$, by direct calculation, we can obtain
(3)$$D_{KL}\left(P_{1}∥P_{2}\right)=\frac{1}{2}\left\{\log\frac{\left|\Sigma_{2}\right|}{\left|\Sigma_{1}\right|}-n+\mathrm{tr}\left(\Sigma_{2}^{-1}\Sigma_{1}\right)+{\left(\mu_{2}-\mu_{1}\right)}^{T}\Sigma_{2}^{-1}\left(\mu_{2}-\mu_{1}\right)\right\}.$$ Under the parameter coordinate (μ,Σ), the expression of the geometric mean $c\left(C\right)=\underset{P\in N_{n}}{\mathrm{argmin}}\frac{1}{m}\sum_{i=1}^{m}D_{KL}\left(P_{i}\left|\right|P\right)$ is very complicated, and it is not convenient to use. In order to overcome the difficulty, we will throughout the equation change the probability density function of $P\in N_{n}$ into the form of exponential distribution. In fact, by setting $x_{1}=x$, $x_{2}=-\frac{1}{2}x^{T}x$ and $\theta_{1}=\Sigma^{-1}\mu ,\theta_{2}=\Sigma^{-1}$, we can obtain the form of exponential distribution
(4)$$P\left(x;\mu ,\Sigma \right)=P\left(x_{1},x_{2};\theta \right)=\exp\left\{\langle \bar{x},\theta \rangle -\varphi \left(\theta \right)\right\},$$
where $\bar{x}=\left(x_{1},x_{2}\right)$, $\theta =\left(\theta_{1},\theta_{2}\right)$ is called the natural parameter, $\langle \bar{x},\theta \rangle$ is the inner product of $\bar{x}$ and θ, and the function $\varphi \left(\theta \right)=\frac{1}{2}\left(\theta_{1}^{T}\theta_{2}^{-1}\theta_{1}-\log\left|\theta_{2}\right|-n\log2\pi \right)$ is called the potential function, which is a convex function.

By using the potential function φ, we can define the generalized KL divergence, namely the Bregman divergence on $N_{n}$, as
(5)$$B_{\varphi }\left(P_{2}∥P_{1}\right):=\varphi \left(\theta_{2}\right)-\varphi \left(\theta_{1}\right)-\langle \nabla \varphi \left(\theta_{1}\right),\theta_{2}-\theta_{1}\rangle ,$$
where $\theta_{1},\theta_{2}$ are two parameters of $N_{n}$.

**Remark** **1.**
*By means of the exponential form for the probability density functions $P_{1},P_{2}\in N_{n}$, direct calculation yields*

$$B_{\varphi }\left(P_{2}∥P_{1}\right)=D_{KL}\left(P_{1}∥P_{2}\right).$$



### 2.3. Mean of Parameter Point Clouds

The main idea of the traditional K-means algorithm is that for a given data cloud with the scale *m*,
$$C_{m}=\left\{p_{i}\in R^{n}|i=1,\cdots ,m\right\},$$
which is abbreviated as *C*, by using the clustering algorithm, we divide the point cloud into *K* classes. The effect of the traditional K-means algorithm is mainly affected by the selection of initial cluster centers, the expression of data and the difference function.

In order to avoid the shortage of the traditional K-means algorithm, we will consider the clustering algorithm on the Riemannian space instead of the Euclidean space so that we can use the geodesic distance and KL divergence but the Euclidean distance and obtain better clustering results.

Now, we give the definition of the geometric mean of point cloud *C* in $N_{n}$ under different difference functions *D*.

**Definition** **5.**
*The geometric mean c(C) of point cloud $C=\left\{\left(\mu_{1},\Sigma_{1}\right),\cdots ,\left(\mu_{m},\Sigma_{m}\right)\right\}$ in $N_{n}$ is*

$$\begin{array}{c} \begin{array}{c} c\left(C\right):=\underset{\left(\mu ,\Sigma \right)\in N_{n}}{\mathrm{argmin}}\frac{1}{m}\sum_{i=1}^{m}D\left(\left(\mu_{i},\Sigma_{i}\right),\left(\mu ,\Sigma \right)\right). \end{array} \end{array}$$



In practical problems, the calculation of the geometric mean of some difference functions may be very complicated; thus, we will use the arithmetic mean instead of the geometric mean.

**Definition** **6.**
*The parameter space $R^{n}\times \mathrm{SPD}\left(n\right)$ of $N_{n}$ is a convex set. Hence, the arithmetic mean $\bar{c}\left(C\right)$ of the parameter point cloud $C=\left\{\left(\mu_{1},\Sigma_{1}\right),\cdots ,\left(\mu_{m},\Sigma_{m}\right)\right\}$ in $N_{n}$ can be defined as   *

$$\bar{c}\left(\tilde{C}\right)=\frac{1}{m}\sum_{i=1}^{m}\left(\mu_{i},\Sigma_{i}\right).$$



Now, we introduce the geometric mean of the point cloud *C* with respect to the KL divergence.

From (Equation 5), we can obtain the following proposition [16].

**Proposition** **2.**
*The geometric mean of the point cloud C with respect to the KL divergence exists and is unique, and is equal to the arithmetic mean in the above natural coordinates.*


Furthermore, we can see that the geometric mean of the point cloud *C* with respect to the Bregman divergence $B_{\varphi }$ exists and is unique, and it is equal to the arithmetic mean in natural coordinates, hence the geometric mean of point cloud *C* about KL divergence exists and is unique, and it is equal to the arithmetic mean in natural coordinates, that is,
(6)$$c\left(C\right)=\underset{P\in N_{n}}{\mathrm{argmin}}\frac{1}{m}\sum_{i=1}^{m}D_{KL}\left(P∥P_{i}\right)=P\left(x_{1},x_{2};\frac{1}{m}\sum_{i=1}^{m}\theta_{i}\right).$$

In the following K-means algorithm with KL divergence as the difference function, the Proposition 2 ensures that the geometric mean of the parameter point cloud can be explicitly given by the arithmetic mean after parameter transformation.

### 2.4. Gaussian Mixture Models

The mixture model is a probability model that can be used to represent an overall distribution with K sub-distributions. In other words, the mixture model represents the probability distribution of observational data overall, which is a mixture of K sub-distributions. The mixture model does not require the observational data to provide information about the sub-distributions to calculate the probability that the observational data are in the overall distribution.

In general, a mixture model can use any probability distribution, but due to the good mathematical properties and good computational performance of the Gaussian distribution, the Gaussian mixture model is the most widely used model in practice [17].

**Definition** **7.**
*The probability distribution of Gaussian mixture models is*

(7)
$$P\left(x|\Theta \right)=\sum_{i=1}^{K}\alpha_{i}p_{i}\left(x|\theta_{i}\right),$$

*where $\Theta =\left(\alpha_{1},\ldots ,\alpha_{K},\theta_{1},\ldots ,\theta_{K}\right)$ such that $\alpha_{i}\geq 0,\;\sum_{i=1}^{K}\alpha_{i}=1$, $\alpha_{i}$ is the probability that the observational data belong to the i-th submodel and $p_{i}$ is the Gaussian distribution density function of the i-th submodel, whose parameter is $\theta_{i}$.*


## 3. Statistical K-Means Algorithm

The K-means algorithm on statistical manifolds, which we refer to as the SKM algorithm, consists of three parts: local statistical method, K-means algorithm, and selection of difference function. This section first introduces the K-nearest neighbor local statistical method and then introduces the details of the SKM algorithm.

### 3.1. Local Statistical Method

The point cloud is a sampling of some specified features in the objective world, each of which we consider to have the same properties within a small neighborhood. Mathematically, we obtain neighborhood properties through local statistics. Specifically, we use local statistics as parameters to describe a parameter distribution. Two sets of different local statistics can determine two different distributions on the same parameter distribution family. This idea is equivalent to finding a distribution for any point in the point cloud and its neighbors in the point cloud (subclouds of the point cloud) such that the subcloud is a sample of that distribution.

For the initial point cloud without any annotation, we have no reason to think that its local statistics conform to some special distribution. We believe that the factors affecting the local distribution of point clouds in their natural background are complex enough; consequently, the local statistics can be generated from a multivariate normal distribution according to the Central Limit Theorem. Therefore, we only need to calculate the mean and covariance matrix of each point of the point cloud in its local area to determine a normal distribution. By doing this, the entire point cloud will be projected as a parameter point cloud on the family of multivariate normal distribution, and then, the K-means algorithm is used on the parameter point cloud to cluster the original data. The data are then classified using their differences in neighborhood densities [18,19,20,21].

For the selection of the neighborhood in the point cloud, we use the *k*-nearest neighbor method: that is, for any positive integer *k*, find a *k* Euclidean nearest neighbor of some point in the point cloud. This method can reflect the number density of local point clouds. Next, we introduce the selection method of k-nearest neighbors.

**Definition** **8.**
*Let $C_{m}=\left\{p_{i}\in R^{n}|i=1,2,\cdots ,m\right\}$ be a point cloud of scale m, abbreviated C. For any $p\in C_{m}$,*

$$k-N\left(p,k\right)=\left\{p_{j}\in C_{m},j\in \left[i_{1},\cdots ,i_{k}\right]|∥p_{l}-p∥\geq ∥p_{j}-p∥,\forall l\notin \left[i_{1},\cdots ,i_{k}\right]\right\}$$

*is called the k-nearest neighbor of p in $C_{m}$, abbreviated as k-N, and p∈k-N⊆C.*


Denote $\mu \left(k-N\right)=E\left[k-N(p,k)\right]-p$ and $\Sigma \left(k-N\right)=\mathrm{Cov}\left[k-N(p,k)\right]$ as the mean and covariance matrices of the distances between data points in *p* and N(p,k), respectively, thus defining the local statistical map
(8)$$\Psi_{k}:C\to N_{n},$$
where $\Psi_{k}\left(p\right):=f\left(\mu \left(k-N\right),\Sigma \left(k-N\right)\right)=\frac{1}{\sqrt{{\left(2\pi \right)}^{n}\det\left(\Sigma \right)}}\exp\left\{-\frac{{\left(x-\mu \right)}^{T}\Sigma^{-1}\left(x-\mu \right)}{2}\right\}$. It is worth noting that we refer to the image of point cloud *C* under the local statistical map $\Psi_{k}$
$$\tilde{kC}:=\Psi_{k}\left[C\right],\tilde{kC}\subseteq N_{n}$$
as the parameter point cloud under the k-nearest neighbor method in this paper.

### 3.2. Details of the SKM Algorithm

Giving the image of point cloud *C* under the k-nearest neighbor and local statistical mapping $\tilde{kC}=\Psi_{k}\left[C\right]$, which is the parameter point cloud in $N_{n}$, it is reasonable that we cluster the parameter points to gain the potential classifications among the original data, and the core idea of the SKM algorithm is the application of the K-means algorithm together with non-Euclidean difference functions. The SKM algorithm’s performance depends on the choice of difference functions, which makes the SKM algorithm flexible for various tasks.

The specific steps of the SKM algorithm are as Algorithm 1:
**Algorithm 1**  Statistical K-Means Cluster Algorithm**Input:** point cloud *C*, k-nearest neighbor indicator *k*, initial cluster center $c_{1}^{0},\cdots ,c_{k}^{0}$, threshold ε**Output:** a K division of point cloud *C*1: By local statistics methods, the point cloud *C* is represented as a point cloud in the manifold of *n*-dimensional normal distribution family $\tilde{kC}$2:Input the initial cluster centers $c_{1}^{0},\cdots ,c_{k}^{0}$ and, based on the selected difference function, apply the K-means algorithm to $\tilde{kC}$, where the distances between parameter points are given by the difference function, and the centroid $c_{j}^{i}$ is updated to the current geometric mean of each division3:According to the indicator division of $\tilde{kC}$ clustering $l_{1},\cdots ,l_{k}$, the output $C\left[l_{1}\right],\cdots ,C\left[l_{k}\right]$ is a division of the origin cloud *C*


## 4. Data Pre-Processing and Preparations

After the introduction of the SKM algorithm, we can prepare the data for our method to simulate on. This section mainly explains the work of data pre-processing and the criteria to assess the cluster results.

### 4.1. Data Pre-Processing

Here, the original data of the experiment are selected among the top 20 universities in mainland China in terms of scientific research funding in 2021. A total of 32 types of indicators from 2010 to 2019 are taken into account. Data sources are the WOS and CSSCI databases alongside the analysis platform of CNKI [22,23,24]. The names of universities and statistical indicators are as Table 1 and Table 2.

Assuming that $x_{i}$ as the *i*-th indicator, the numerical expression of academic performance of a university *s* in the year *y* is denoted by
$$X_{s,y}={\left(x_{1},x_{2},\cdots ,x_{k}\right)}^{T}.$$

It is natural that we make up a matrix X(s,y) whose element is the academic performance vector $X_{s,y}$. Hence, the row represents different universities, and the column represents the different years. Since our indicators are in different dimensions, we apply the z-score normalization on the indicators of every column: namely, normalize the same indicator of different universities in the year.
$$x_{nor}=\frac{x-mean(X)}{std(X)},\;x\in X.$$

The normalization makes indicators among different years comparable, which forms the basis of clustering.

### 4.2. Clustering Assessment Criteria

The commonly used clustering assessment criteria can be generally devided into two classes, external assessment and internal assessment. The external assessment needs a reference model as the benchmark, while the internal assessment simply measures the clustering results from the perspective of compactness, connectivity and so on. Since there is no state-of-the-art reference model or ranking in this field, it is convincing to choose proper internal assessment criteria. In this paper, we use the Davies–Bouldin Index (DBI), Dunn Index (DI) and Silhouette Score (SC) as the clustering assessment criteria, which have been proved to be effective in such problems [25,26].

Assume that $C=\{C_{1},C_{2},\cdots ,C_{k}\}$ as the cluster result, where |C| represents the number of samples in *C*, $\mathrm{dist}(x_{i},x_{j})$ represents the distance metric of sample $x_{i}$ and $x_{j}$, $\mu_{i}$ represents the center of cluster $C_{i}$. Giving definitions as follows
$$\mathrm{avg}\left(C\right)=\frac{2}{\left|C|(|C|-1\right)}\sum_{1\leq i\leq j\leq |C|}\mathrm{dist}\left(x_{i},x_{j}\right),$$
$$\mathrm{diam}\left(C\right)=\underset{1\leq i\leq j\leq |C|}{max}\left\{\mathrm{dist}\left(x_{i},x_{j}\right)\right\},$$
$$d_{min}\left(C_{i},C_{j}\right)=\underset{x_{i}\in C_{i},x_{j}\in C_{j}}{min}\left\{\mathrm{dist}\left(x_{i},x_{j}\right)\right\},$$
$$d_{cen}\left(C_{i},C_{j}\right)=\mathrm{dist}\left(\mu_{i},\mu_{j}\right).$$

Then, we can define DBI, DI and SC as
$$DBI=\frac{1}{k}\sum_{i=1}^{k}\underset{i\neq j}{max}\left(\frac{\mathrm{avg}\left(C_{i}\right)+\mathrm{avg}\left(C_{j}\right)}{d_{cen}\left(\mu_{i},\mu_{j}\right)}\right),$$
$$DI=\frac{\underset{1\leq i\leq j\leq m}{min}\left\{d_{min}\left(C_{i},C_{j}\right)\right\}}{\underset{1\leq l\leq m}{max}\left\{\mathrm{diam}\left(C_{l}\right)\right\}},$$
$$s\left(x_{i}\right)=\frac{b-a}{max(a,b)},\;a=\frac{1}{|C_{q}|-1}\sum_{x_{i},x_{j}\in C_{q}}\mathrm{dist}\left(x_{i},x_{j}\right),$$
$$b=\frac{1}{\sum |C|-|C_{q}|}\sum_{x_{i}\in C_{q},x_{j}\notin C_{q}}\mathrm{dist}\left(x_{i},x_{j}\right),\;SC=\frac{\sum s(x_{i})}{\sum |C|}.$$

The three indicators evaluate the clustering results from different perspectives. DBI measures the maximum similarity between clusters; hence, the smaller DBI is, the better the clustering result is; DI calculates the ratio of the minimum cluster distance and the largest intra-class discrete distance, and a good clustering result should make the value as big as possible; the SC value of each sample represents the degree of matching relationship between the sample and its cluster; therefore, the higher the SC value in general, the better the clustering result.

## 5. Data Cloud Simulation

In this section, we will respectively apply the traditional K-means, GMM and the SKM algorithm on the processed data. By analyzing the cluster results and calculating the assessment criteria scores, we can compare the performance of different algorithms as well as give the academic levels of the 20 universities. The estimation of university academic level is given by the most reasonable cluster result, as all these cluster algorithms evolve random processes.

### 5.1. The K-Means Algorithm Clustering

To avoid the influence of sparse data and speed up the process of convergence, PCA is used at first to reduce the data dimension [27]. The PCA scree plot is displayed as Figure 1.

Often, there are two ways to obtain the number of principal components, that is, to retain a certain percentage of the variance of the original data or to retain only the principal components with eigenvalues greater than 1 according to Kraiser’s rule [28,29]. It can be seen in the shown PCA results that there are five principal components with eigenvalues greater than 1, and when the number of principal components is 6, the cumulative variance contribution rate reaches more than 0.8. We finally choose to keep six principal components, that is, compress the 32-dimension original data to six dimensions. It is worth mentioning that several indicators ignored in previous research prove to contribute signficantly according to the PCA results, which are shown above. This is a strong testament to the effectiveness of big data.

There are many methods for deciding the number of clusters *K*. One simple way is to observe the sum of the squarred errors (SSE) with the change of *K* and select the point where SSE changes from steep to gentle. However, the Figure 2 shows that there is no very clear elbow point. As a consequence, we choose to use the Gap Statistic method [30]. Every *K* corresponds to a $Gap_{k}$ and $s_{k}$, and *K* is selected as the minimal *K* that makes $Gap_{k}-Gap_{k+1}+s_{k+1}\geq 0$. We conduct simulations 50 times, as random sampling is also used in the Gap Statistic. The results are shown in Figure 3, and Figure 4 shows the most common case. It can be seen that when K=4,6, the GapDiffs are most likely to be greater than 0. Although inferior to K=4,6, K=5 also shows a considerable frequency. Considering that academic performance evaluation needs an adequate *K* to produce reasonable results, we finally chose *K* as 4, 5 and 6.

In order to obtain credible results, we limit the iteration times of each simulation to 20, so as to avoid bad cases caused by random initialization. In addition, we merge those simulations that have very similar initialization and cluster results. We select the most representive case by comparing their clustering evaluation criteria [31,32]. This strategy makes it easier for us to analyze the performance of different algorithms. For eack *K*, we conduct 30 independent simulations and give the cluster details. To better visualize the clusters, we map the original data points to a plane using PCA. The results are shown in the table and graph below.

When K=4, we can see from Figure 5 that the cluster completeness is well preserved. Only Xi’an Jiaotong University and Tongji University have small parts divided into different clusters, and the rest of the data points of the same university are all in the same cluster.

When K=5, the cluster result Figure 6 still shows very good completeness. However, some universities have changed from one cluster to another. Peking University itself becomes one new cluster, and Wuhan University becomes clustered with Beijing Normal University and Fudan University.

When K=6, things begin to change. We can see from Figure 7 that so-called rag bags, which mean small parts of data points that cannot be well clustered, begin to increase. This actually has a bad effect on the cluster homogenity. Shanghai Jiao Tong University and Sun Yat-sen University now also change the cluster and join with Fudan University, while Wuhan University and Beijing Normal University remain together.

It can be seen from Table 3 that the clustering indicators of the K-means algorithm are relatively stable. DBI is basically maintained between 1.3 and 1.6, DI is basically maintained between 0.05 and 0.08, and SC is mostly distributed above 0.3. It is in line with the previous SSE result and proves the cluster result to be reasonable. For results, with the change of *K*, the data points of Tsinghua University and Zhejiang University in all years are always the only two in the same cluster, which indicates that the academic level of these two universities is very close and there is a large gap between the two and the remaining universities. In addition, in all years data points of Central South University, Jilin University, Sichuan University, Huazhong University of Science and Technology, Shandong University, Tongji University, etc. always appear in the same cluster, indicating their academic level is close; Northwestern Polytechnical University, Beihang University, Beijing Institute of Technology, Harbin Institute of Technology and Southeast University are in the same situation, and the difference between these two clusters may be that the universities in the latter cluster have a strong color of science and engineering along with a national defense background. Considering that Xi’an Jiaotong University has a relatively uniform distribution in the two clusters with the change of *K*, it is likely that the academic level is close. We also notice that the clustering results of Wuhan University, Sun Yat-sen University, Fudan University, Shanghai Jiao Tong University, Beijing Normal University, Peking University and other universites changed greatly with the change of *K*. When K=4, Beijing Normal University and Peking University are in the same cluster, but it is then divided as *K* increases. One explanation is that when *K* is small, Beijing Normal University and Peking University are clustered together because they have similar backgrounds in humanities and social sciences. However, because of the huge difference of academic level, the two are then divided. This also explains the cluster variance for Fudan University, Sun Yat-sen University, Wuhan University, and Shanghai Jiao Tong University. These are all comprehensive universities, and characteristics of both (1) humanities and social science and (2) science and engineering are relatively distinct. Therefore, for different *K*, they can be in the same cluster with Beijing Normal University or in the cluster of science and engineering backgrounds.

### 5.2. The GMM Clustering

Different from the K-means algorithm, the Gaussian mixture model uses Gaussian distributions as feature descriptors, and it is able to softly assign weights for each component thanks to the Expectation Maximization (EM) algorithm. Consequently, the GMM can form clusters of more complicated shapes, which makes it suitable for the university academic data. Under the consideration of consistence with K-means and from the experience of previous work [33], we take the same simulation conditions as the K-means. The Gap Statistic method can also be applied to the GMM, so it is reasonable to choose the same *K* values. The results are shown in the table and graph below.

We can see from Table 4 that the overall performance of the GMM is better than the K-means in terms of clustering criteria. During the change of N-class, we can see that there are actually two patterns. The results of Figure 8 and Figure 9 are actually very similar to that of the K-means. However, Figure 10 and Figure 11 present a very unbalanced result. In thier case, almost all the universities of science and technology are clustered together, and the rest of the universites are actually always the same ones. Although good cluster criteria scores are obtained, the results of the GMM actually cannot be used for university academic evaluation, as they make no effective divisions. This indicates that a different feature extraction method is needed, and we use the SKM algorithm.

### 5.3. The SKM Algorithm Clustering

The idea of the SKM algorithm is based on the assumption that in the original data point cloud, the neighborhood of each point should have a convergent property with this point. The point cloud is the sampling and discretization of real physical quantities, so the rationality of this assumption is quite natural. In our simulation, we firstly use the k-nearest neighbor method to select points near each data point and map this subcloud to an N-dimensional normal distribution family manifold. Then, we apply the SKM algorithm with non-Euclidean difference functions and analyze their clustering results. For the selection of *k*, we simply choose k=10, which is the number of the points in the origin point cloud for every university. The choice not only enables the points from the same university to be mapped to one distribution on statistical manifolds in theory: it also has been proven in our simulation that when k=10, the SKM algorithm could achieve convergence faster compared to other *k*-values.

In this simulation, we use the KL divergence and the Wasserstein difference functions. Due to the use of the local statistical method, there is no need for dimension reduction; in other words, the application of PCA is skipped. Especially, as there is a one-to-one correspondence between the point clouds on Euclidean space and on manifolds, and in the Euclidean space we have obtained *K* values, we just keep it unchanged as our simulation parameters [34]. The other simulation strategies are the same as those in Section 4.1. The results are shown in the table and graph below.

The first is the result of using KL divergence.

When K=4, we can see similar results with K-means from Figure 12; the cluster completeness is also well preserved. However, this time, Peking University is divided into a separate cluster, and Beijing Normal University is divided into a large cluster.

Compared with K-means, we can see from Figure 13 that the biggest difference when K=5 is that this time, Sun Yat-sen University, Fudan University, and Shanghai Jiao Tong University are in the same cluster. Except for Peking University, Zhejiang University, and Tsinghua University, the rest are divided into two main clusters.

When K=6, it also fails to cluster a small number of data points well. In Figure 14, Peking University, Tsinghua University, and Zhejiang University were each divided into a cluster.

The result for the Wasserstein distance is below.

When K=4, we can see from Figure 15 that the difference between using Wasserstein distance and KL divergence is that when using Wasserstein distance, Fudan University is divided into the same cluster as Peking University. The rest of the results are basically the same.

When K=5, the SKM results in Figure 16 are basically the same with using Wasserstein distance and KL divergence, but with using KL divergence, it is more likely that small parts of data points cannot be well clustered.

When K=6, the clustering results with using Wasserstein distance in Figure 17 are less stable relative to KL divergence. In addition, the Wasserstein distance produce clusters with a very small number of samples, which indicates that it cannot distinguish the mainfolds on this problem very well.

We can see from Table 5 and Table 6 that the SKM algorithm is inferior to the K-means and GMM method on the two indicators of DBI and DI. From the definitions of DBI and DI, we speculate that this can be caused by the local statistical methods. During the process of selecting a local point cloud, we use the K-nearest neighbor strategy. It can better reflect the statistical density characteristics of a local point cloud, but on the other hand, it may also cause the selected area to be non-convex, resulting in a diffrent distribution in parameter space from the original space. However, the SC indicator of both metrics for the SKM algorithm performs better than that in K-means and GMM. We attribute this to the introduction of non-Euclidean metrics, which achieve a more granular comparison. It can also be seen from the degree of dispersion of the statistical indicators that the two indicators in this section fluctuate considerably, as the selection of the initial cluster center will greatly affect the final clustering, which is a manifestation of the high sensitivity of the SKM algorithm. Between the two metric functions of the SKM algorithm, the KL divergence performs better, as it gives more stable results and better interpretability, while the Wasserstein distance has greatly varied indicators and gives clusters of high similarities.

In terms of clustering results, the clusters given by the SKM algorithm are generally similar to the results of K-means and general cases of GMM, and they actually have better discrimination on the universities of science and technology than the other case of GMM, but there are still some interesting phenomena. After verification and comparison, it can be seen that using several Riemann metrics defined on symmetric positive definite manifolds, the obtained clustering effect is not as good as KL divergence. Hence, we choose KL divergence as the distance function for clustering. In the results of KL divergence, the clustering results are relatively more stable and have no university spans from one cluster to another. The biggest difference is that the KL divergence does not give a division among comprehensive universities; instead, it further divides universities of science and engineering, resulting in the cluster of Peking University, Beihang University and Northwestern Polytechnical University as well as Harbin Institute of Technology, Southeast University, Xi’an Jiaotong University. As for Wasserstein distance, it has unsatifactory indicators and results. Especially when K=6, the Wasserstein metric produce clusters with a very small number of samples, which indicates that it cannot distinguish the mainfolds on this problem very well. It is worth noting that the dimension of the data on which the SKM algorithm is applied is 32 compared to six for the traditional K-means and GMM algorithms. In this case, the SKM algorithm still obtains remarkable clustering results, which proves the potential of the SKM algorithm in terms of processing large amounts of high-dimensional data.

To further assess the three algorithms quantitatively, we apply them on a UCI ML dataset [35] and compare the accuracies. We choose to use the ’Steel Plates Faults Data Set’ provided by Semeion from the Research Center of Sciences of Communication, Via Sersale 117, 00128, Rome, Italy. Every sample in the dataset consists of 27 features, and the task is to classify whether a sample has any of the seven faults. We choose this dataset because it has similar feature dimensions with our origin problem and it provides various indicators to classify, which can better assess the different clustering algorithms. The results are produced under the same condition as the simulation set above, including data pre-processing methods and cluster parameters. The classification accuracies of different algorithms on the seven faults are shown in Table 7.

We can see that the SKM algorithm is greatly advantageous over the K-means and the GMM algorithm on accuracy scores. In comparison, the dataset provider’s model has an average accuracy of 0.77 on this dataset [36]. In addition, in terms of cluster indicators, we can see from Table 8 that the SKM algorithm has better performance on the SC score, but it does not perform well on the DBI score, which is basically consistent with the results on the Chinese University dataset. The result exactly reveals the great potential of the SKM algorithm on the application of many other fields. It could be a great replacement of traditional Euclidean-based cluster methods in a certain problem.

## 6. Conclusions and Future Work

In this paper, we propose a university academic evaluation method based on statistical manifold combined with the K-means algorithm, which quantifies the academic achievement indicators of universities into point clouds and performs clustering on Euclidean space and the family of multivariate normal distributions manifolds, respectively. The simulation results show that in terms of DBI and DI, the SKM algorithm is inferior to the method of direct PCA weight reduction and K-means clustering in Euclidean space. On the SC indicator, the SKM algorithm is significantly better than the traditional K-means method in both difference functions. The GMM has a slightly better performance than the K-means, but it still lacks necessary discrimination to tell apart the universites of similar backgrounds. This shows that the SKM algorithm can extract features that are hard to capture in Euclidean space, thus achieving more fine-grained feature recognition and clustering. The great ability is attributed to the process of mapping original data to the local statistics, which forms the parameter distribution on statistical manifold.

By analyzing the cluster results, we can also demonstrate that most of the universities evaluated have very similar academic levels, and their main differences come from their developing backgrounds. This conclusion explains the reason why university ratings could vary greatly in different leaderboards, and it indicates that different evaluation perspectives may be taken for different universites. Clustering would be useful when seperating different types of universities, and this paper provides a promising way.

In the future, we need to strictly construct the theoretical model of the point cloud and explain the principle of local statistics according to the theory of probability theory. On this basis, we try to propose other local statistical methods and analyze their effectiveness. Furthermore, this paper discusses the case where KL divergence and Wasserstein distance are used as difference functions, and other distance functions can be discussed as difference functions later, which may lead to better clustering algorithms. Finally, the explicit expression of the geometric mean of the Wasserstein distance adopted in this paper is still an unsolved problem, and we replace its geometric mean with the arithmetic mean. If this problem is solved, it is possible that the simulation results of the algorithm will be more accurate.

## Figures and Tables

**Figure 1 entropy-24-01004-f001:**
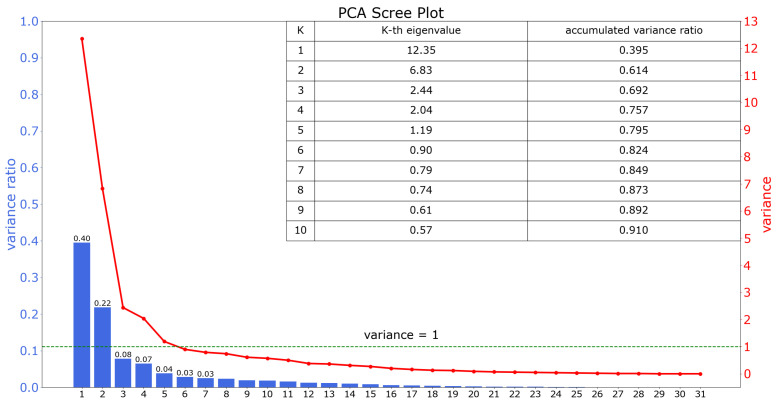
PCA Scree Plot. The red line is the variance plot and explains the proportion of variation by each component from PCA; the green dotted line is the split line to better present components that have variance bigger than 1.

**Figure 2 entropy-24-01004-f002:**
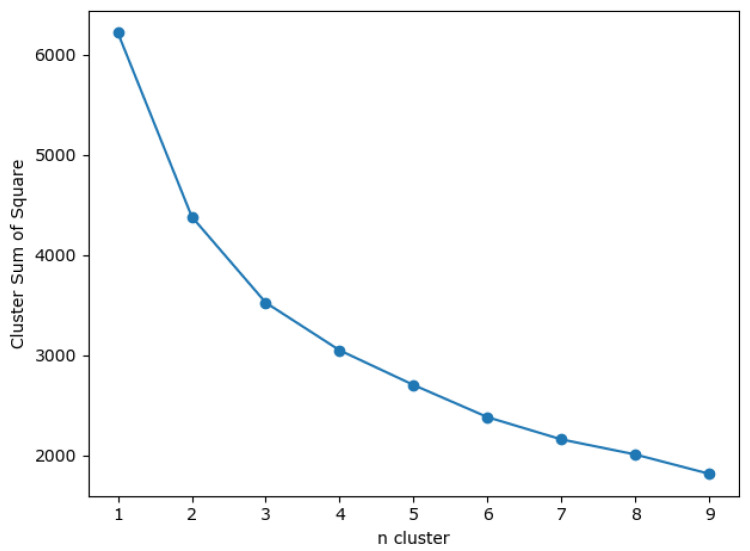
Sum of the Squared Errors Plot.

**Figure 3 entropy-24-01004-f003:**
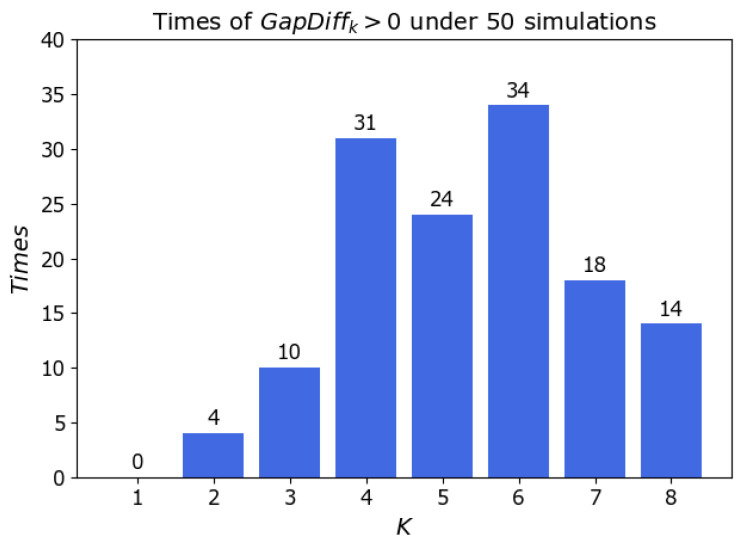
Results of Gap Statistic Simulations.

**Figure 4 entropy-24-01004-f004:**
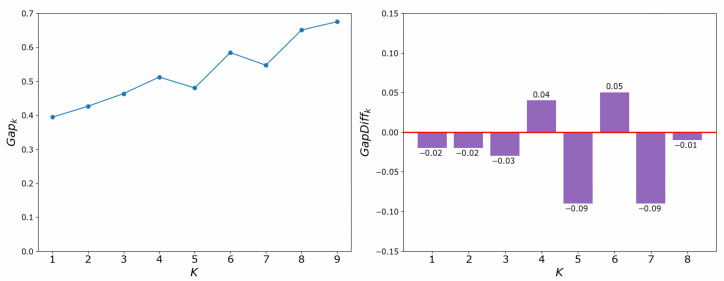
Gap Statistic Typical Result.

**Figure 5 entropy-24-01004-f005:**
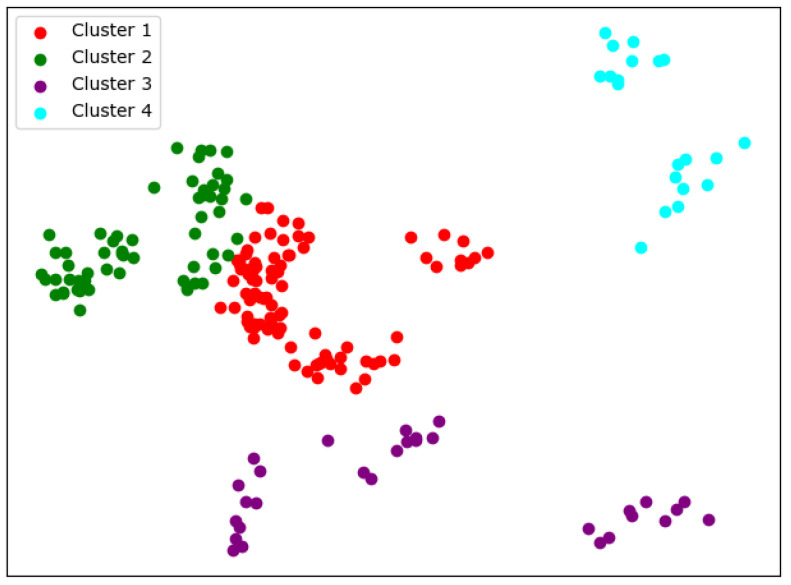
Clustering results of K-means when *K* = 4.

**Figure 6 entropy-24-01004-f006:**
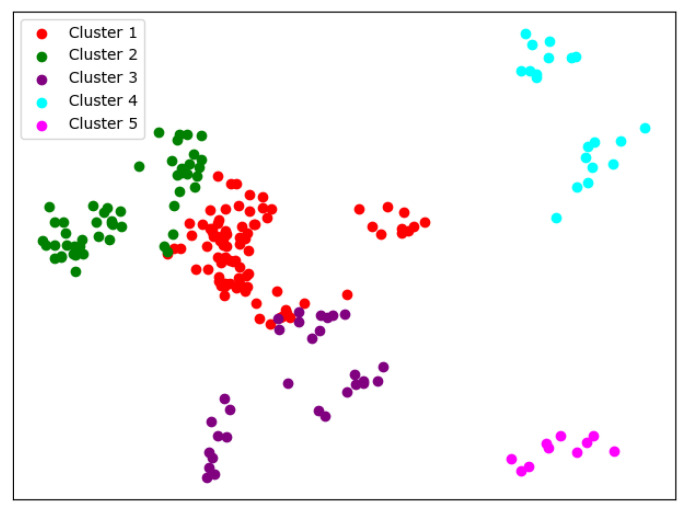
Clustering results of K-means when *K* = 5.

**Figure 7 entropy-24-01004-f007:**
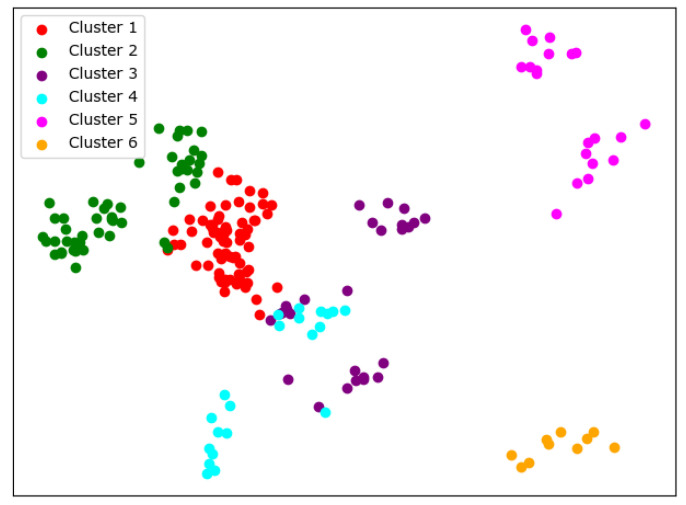
Clustering results of K-means when *K* = 6.

**Figure 8 entropy-24-01004-f008:**
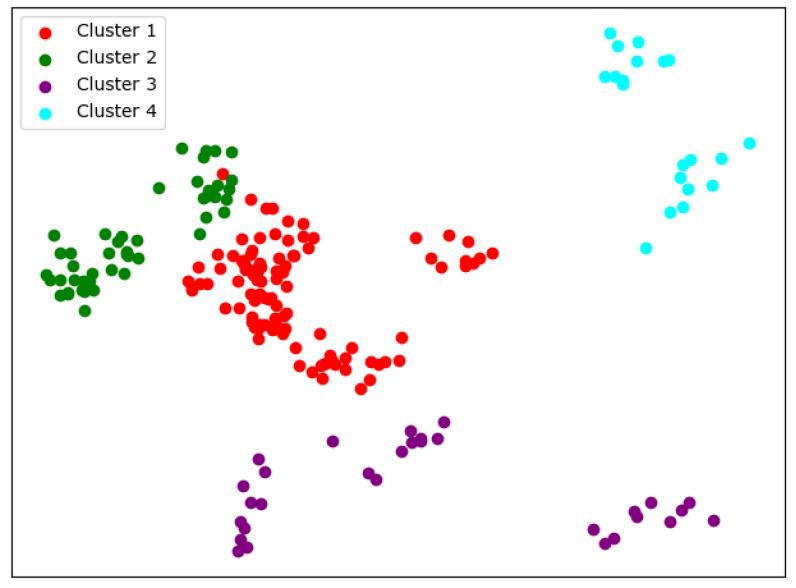
One case of the GMM when N = 4.

**Figure 9 entropy-24-01004-f009:**
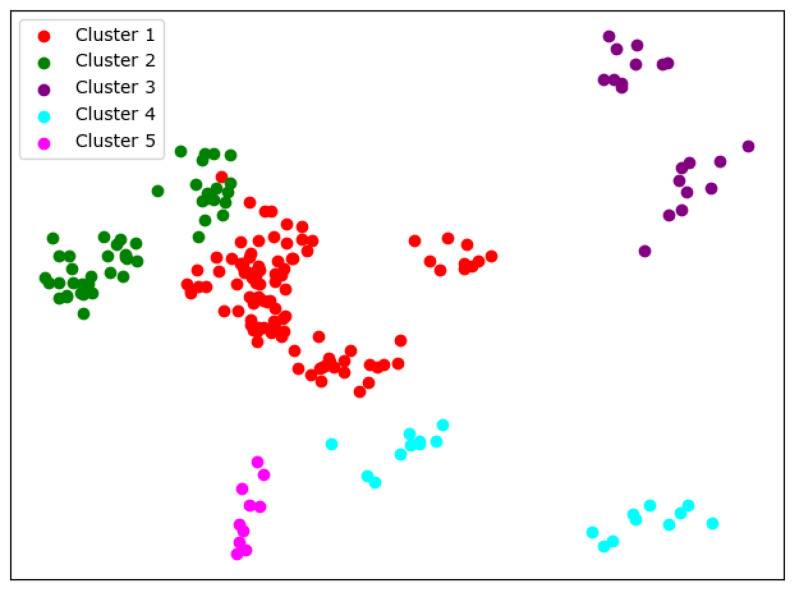
One case of the GMM when N = 5.

**Figure 10 entropy-24-01004-f010:**
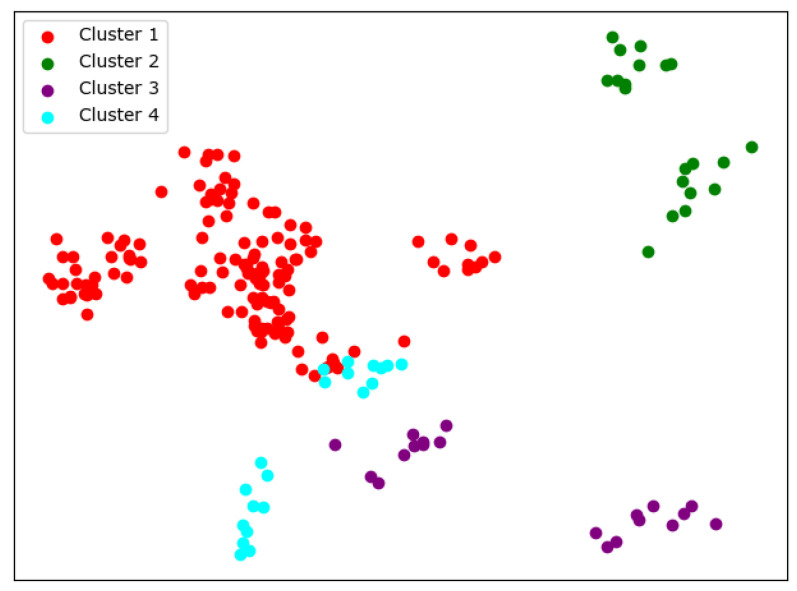
The other case of the GMM when N = 4.

**Figure 11 entropy-24-01004-f011:**
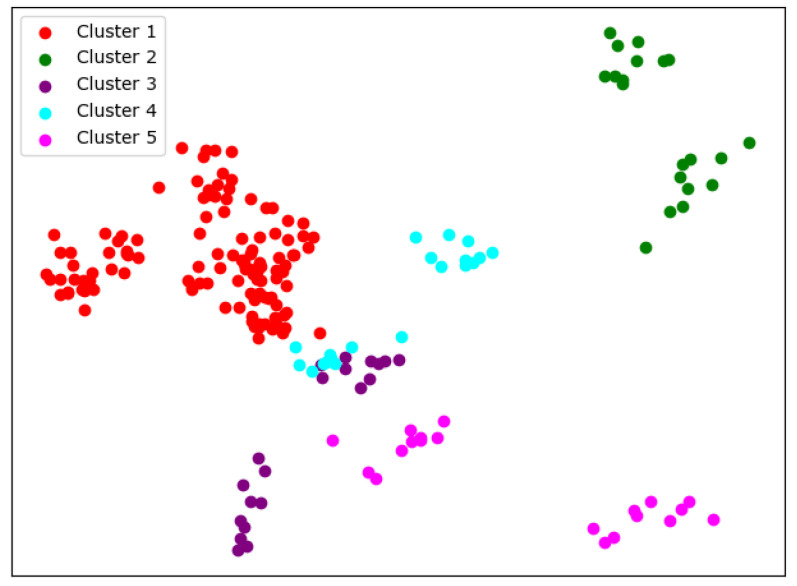
The other case of the GMM when N = 5.

**Figure 12 entropy-24-01004-f012:**
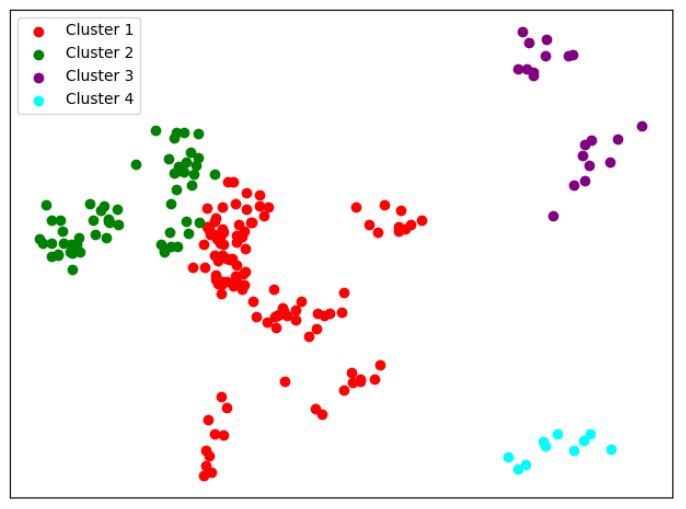
Clustering results of SKM about KL divergence when *K* = 4.

**Figure 13 entropy-24-01004-f013:**
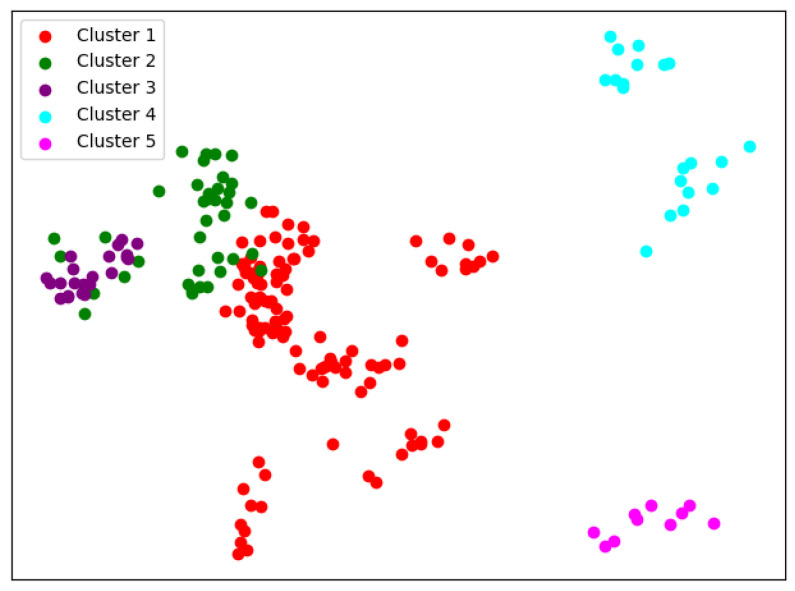
Clustering results of SKM about KL divergence when *K* = 5.

**Figure 14 entropy-24-01004-f014:**
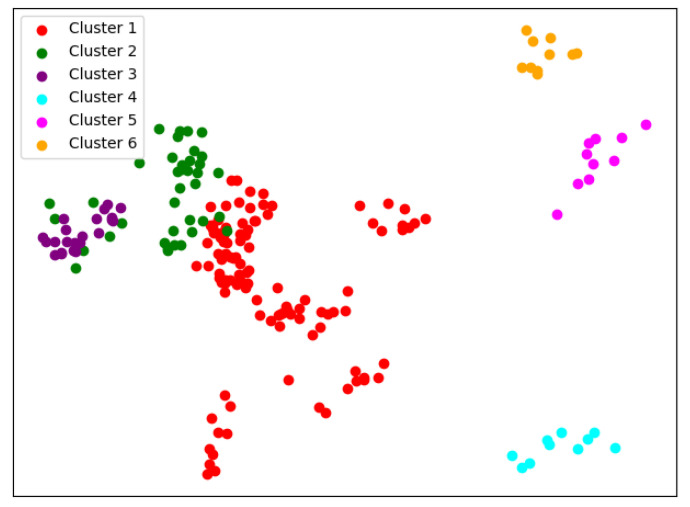
Clustering results of SKM about KL divergence when *K* = 6.

**Figure 15 entropy-24-01004-f015:**
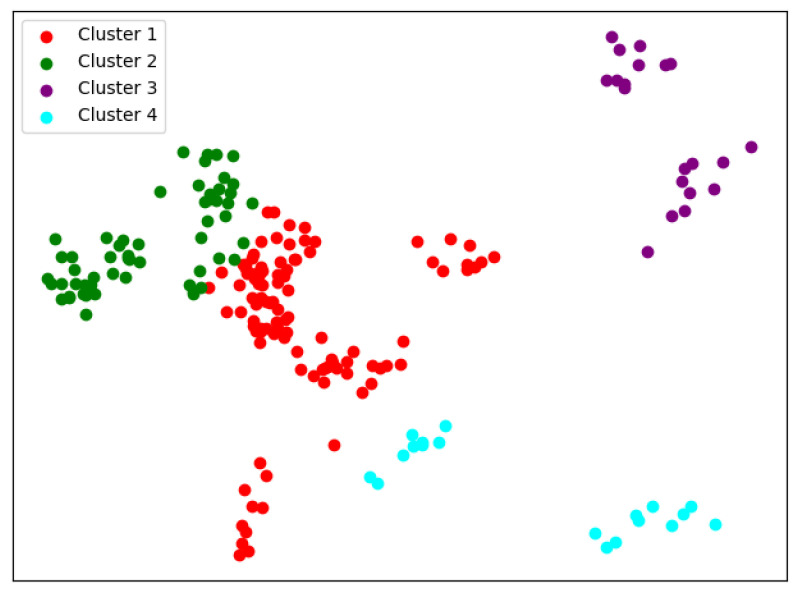
Clustering results of SKM about Wasserstein distance when *K* = 4.

**Figure 16 entropy-24-01004-f016:**
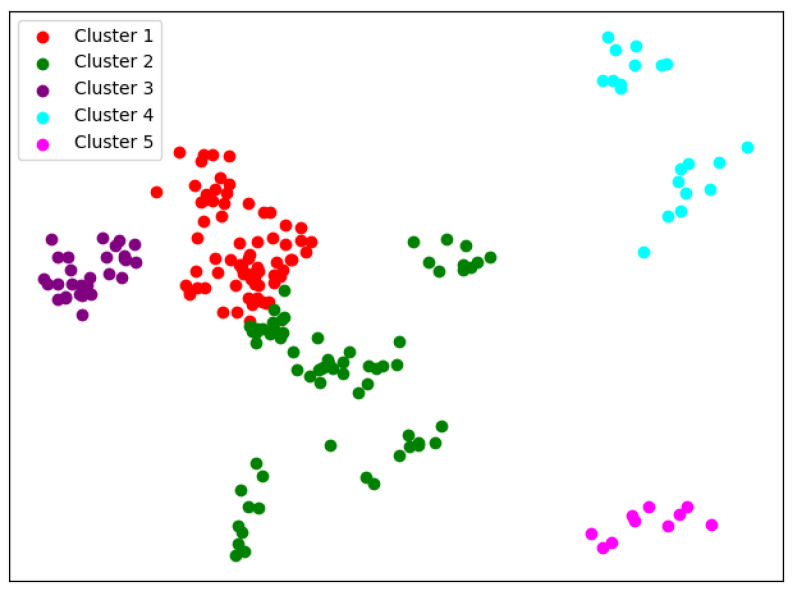
Clustering results of SKM about Wasserstein distance when *K* = 5.

**Figure 17 entropy-24-01004-f017:**
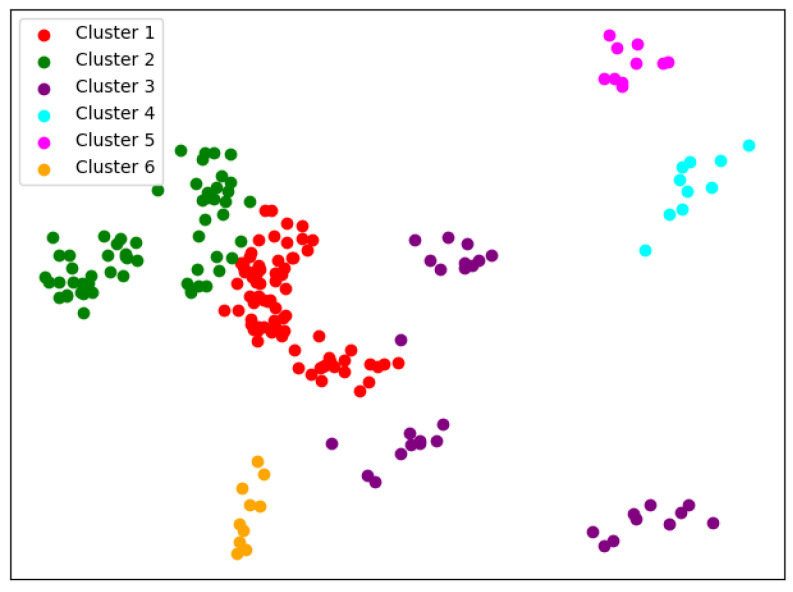
Clustering results of SKM about Wasserstein distance when *K* = 6.

**Table 1 entropy-24-01004-t001:** The names of the twenty universities and their abbreviations.

University Name	Abbreviation
Tsinghua University	THU
Zhejiang University	ZJU
Peking University	PKU
Sun Yat-sen University	SYSU
Shanghai Jiao Tong University	SJU
Fudan University	FDU
Shandong University	SDU
Huazhong University of Science and Technology	HUST
Xi’an Jiaotong University	XJU
Southeast University	SEU
Beihang University	BUAA
Harbin Institute of Technology	HIT
Tongji University	TJU
Wuhan University	WHU
Northwestern Polytechnical University	NPU
Jilin University	JLU
Beijing Normal University	BNU
Central South University	CSU
Beijing Institute of Technology	BIT

**Table 2 entropy-24-01004-t002:** Selection of thirty-two statistical indicators.

Category	Indicator
SCI	Total Posts
	Total Cited
SSCI	Total Posts
	Total Cited
CSCD	Total Posts
	Total Cited
CSSCI	Total Posts
Patent	Number of Patent Applications
	Number of Invention Patent Applications
	Number of Utility Model Patent Applications
	Number of Industrial Design Patent Applications
	Number of Patent Authorizations
	Number of Invention Patent Authorizations
	Number of Utility Model Patent Authorizations
	Number of Industrial Design Patent Authorizations
Funding	Amount of State-Level Funding
	Amount of Ministrial Funding
	Amount of Provincial Funding
	Number of National Natural Science Funds
	Amount of National Natural Science Funding
	Number of National Social Science Funds
Newspaper	Number of Posts
	Number of Citations
	Average Cited
	Number of Downloads
	Average Downloads
	Posts on Local Newspaper
	Posts on Central Newspaper
Rewards	The State Science and Technology Awards
	State-Level Teaching Award
	Honors from Ministry and Province
	Academic Association Awards

**Table 3 entropy-24-01004-t003:** K-means clustering results.

*K*	Number of Cases	Samples in Different Clusters	DBI	DI	SC
4	27	90 60 30 20	1.56	0.06	0.30
4	3	84 65 31 20	1.68	0.06	0.29
5	14	91 50 29 20 10	1.45	0.07	0.33
5	7	86 55 29 20 10	1.47	0.08	0.33
5	4	84 56 30 20 10	1.49	0.07	0.32
5	4	75 67 28 20 10	1.50	0.06	0.32
5	1	82 57 30 20 11	1.40	0.06	0.32
6	10	71 51 27 21 20 10	1.41	0.07	0.34
6	7	74 51 30 20 15 10	1.40	0.07	0.34
6	5	78 51 28 22 11 10	1.39	0.07	0.34
6	4	81 58 20 20 11 10	1.30	0.07	0.35
6	4	78 51 30 20 11 10	1.35	0.07	0.35

**Table 4 entropy-24-01004-t004:** GMM clustering results.

N Class	Number of Cases	Samples in Different Clusters	DBI	DI	SC
4	14	102 48 30 20	1.52	0.08	0.31
4	8	95 75 20 10	1.61	0.08	0.29
4	5	140 20 20 20	1.35	0.18	0.34
4	3	102 48 30 20	1.52	0.08	0.31
5	11	102 48 20 20 10	1.31	0.10	0.33
5	7	91 49 30 20 10	1.46	0.15	0.32
5	6	89 47 30 20 11	1.40	0.06	0.33
5	4	55 53 52 20 20	1.68	0.03	0.27
5	2	120 20 20 20 20	1.34	0.16	0.36
6	11	91 49 20 20 10 10	1.34	0.17	0.33
6	5	83 47 30 20 10 10	1.32	0.07	0.36
6	4	70 51 49 10 10 10	1.50	0.15	0.30
6	4	70 49 40 20 11 10	1.40	0.15	0.33
6	3	118 41 11 10 10 10	1.19	0.14	0.38
6	2	120 20 20 20 10 10	1.19	0.16	0.38
6	1	120 20 20 20 20	1.18	0.21	0.38

**Table 5 entropy-24-01004-t005:** SKM clustering results with KL divergence.

*K*	Number of Cases	Samples in Different Clusters	DBI	DI	SC
4	17	110 60 20 10	2.51	0.04	0.65
4	8	103 67 20 10	2.77	0.04	0.63
4	5	100 60 20 20	3.23	0.03	0.64
5	12	104 46 20 20 10	2.87	0.04	0.66
5	11	109 38 23 20 10	3.40	0.03	0.65
5	4	58 57 55 20 10	3.10	0.04	0.67
5	3	87 52 31 20 10	3.08	0.05	0.66
6	13	109 39 22 10 10 10	3.12	0.04	0.66
6	10	84 43 25 20 18 10	4.55	0.02	0.64
6	7	68 41 39 22 20 10	3.54	0.03	0.69

**Table 6 entropy-24-01004-t006:** SKM clustering results with Wasserstein distance.

*K*	Number of Cases	Samples in Different Clusters	DBI	DI	SC
4	21	119 61 10 10	2.03	0.07	0.54
4	5	140 40 10 10	1.78	0.11	0.58
4	4	101 60 20 19	2.66	0.02	0.54
5	15	101 54 20 19 6	2.34	0.02	0.56
5	8	84 59 37 10 10	2.68	0.04	0.54
5	7	73 67 30 20 10	3.11	0.03	0.55
6	17	101 54 19 10 10 6	2.17	0.04	0.58
6	7	84 59 37 10 9 1	2.37	0.02	0.56
6	6	92 63 19 10 10 6	2.66	0.05	0.56

**Table 7 entropy-24-01004-t007:** Classification Accuracies on the Fault Dataset.

Fault Type	K-Means	GMM	SKM (KL Div.)	SKM (Wass)
Pastry	0.7208	0.7398	0.7450	0.9181
Z-Scratch	0.7084	0.7244	0.7400	0.9016
K-Scatch	0.9366	0.9521	0.9547	0.7991
Stains	0.7609	0.7810	0.7979	0.9624
Dirtiness	0.7697	0.7897	0.7970	0.9711
Bumps	0.5971	0.6131	0.6318	0.7924
Other Faults	0.6033	0.6121	0.6479	0.6528
**Ave. Accu.**	0.7281	0.7433	0.7592	0.8568

**Table 8 entropy-24-01004-t008:** DBI, DI and SC Indicators on the Fault Dataset.

Indicator	K-Means	GMM	SKM (KL Div.)	SKM (Wass)
DBI	1.53	1.43	2.99	2.67
DI	0.01	0.02	0.01	0.02
SC	0.36	0.37	0.54	0.49

## Data Availability

Restrictions apply to the availability of these data. Data was obtained from CNKI and are available at https://usad.cnki.net/, accessed on 12 June 2022 with the permission of CNKI.

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
