# Peer review of "Application of Statistical K-Means Algorithm for University Academic Evaluation"

_entropy, 2022, doi:10.3390/e24071004_

Round 1

Reviewer 1 Report

Authors proposed a university academic evaluation method based on statistical manifold, combined with the K-means algorithm, which quantities the academic achievement indicators of universities into point clouds, and performs clustering on Euclidean space and the family of multivariate normal distributions manifolds respectively. The simulation results showed that in terms of DBI and DI, the SKM algorithm is inferior to the method of direct PCA weight reduction and K-means clustering in Euclidean space. On the SC indicator, the SKM algorithm is significantly better than the traditional K-means method in both difference functions. This shows that the SKM algorithm can extract features that are hard to capture in Euclidean space, thus achieving more fine-grained feature recognition and clustering. The great ability attributes to the process of mapping original data to the local statistics, which forms the parameter distribution on statistical manifold.

The results are correct.

Author Response

Dear Reviewer,

We deeply appreciate the time and efforts you’ve spent in reviewing our manuscript named as “Application of Statistical K-Means Algorithm for University Academic Evaluation” with ID: entropy-1791981. We are extremely grateful for your rating highly of us. Thank you again for your precious time and insightful comments.

Comment: Authors proposed a university academic evaluation method based on statistical manifold, combined with the K-means algorithm, which quantities the academic achievement indicators of universities into point clouds, and performs clustering on Euclidean space and the family of multivariate normal distributions manifolds respectively. The simulation results showed that in terms of DBI and DI, the SKM algorithm is inferior to the method of direct PCA weight reduction and K-means clustering in Euclidean space. On the SC indicator, the SKM algorithm is significantly better than the traditional K-means method in both difference functions. This shows that the SKM algorithm can extract features that are hard to capture in Euclidean space, thus achieving more fine-grained feature recognition and clustering. The great ability attributes to the process of mapping original data to the local statistics, which forms the parameter distribution on statistical manifold.

The results are correct.

Response: We are deeply grateful to the reviewer for the comments.

Reviewer 2 Report

The article proposes a methodology for performing k-means on a Riemannian space.

The idea is interesting but unfortunately there are some points that need to be greatly improved.

1) the introduction is not comprehensive enough and there should be a section on "related works" separately.

2) the section where statistical measures are introduced does not make it clear , except at the, end the main purpose of the work.

3) The results are not significant: it is not clear why k =4,5,6 were chosen even if the elbow rule suggests that the optimal k is k=2.

The elbow rule for the SKM is not reported and it does not make sense to show the comparison for k =4,5,6 for KL and for Wasserstein with classical K-means. This way the results are not meaningful.

4) Again about the results, the images make no sense unless it is explained that those are projections in 2 dimensions of a dataset of 6 dimensions (K-means) and 31 dimensions (SKM) 

5) To validate the procedure one should consider using a known dataset the literature (perhaps using a known repository, e.g. UCI ML) and then use the dataset suggested by the authors.

Author Response

Dear Reviewer,

    We deeply appreciate the time and efforts you’ve spent in reviewing our manuscript named as “Application of Statistical K-Means Algorithm for University Academic Evaluation” with ID: entropy-1791981. According to your suggestions, we have modified our articles and the revises are as follows.

Point 1: The introduction is not comprehensive enough and there should be a section on "related works" separately.

Response 1: We appreciate for your suggestion. We have added more related work and made structural changes in our paper. We have tried our best to reorganize the structure and give a more comprehensive introduction.

Point 2: The section where statistical measures are introduced does not make it clear, except at the end the main purpose of the work.

Response 2: We are sorry that the statistical measures are not explained clearly. We have improved some details in our description and deleted some redundant contents(as another Reviewer suggested). Due to space reasons, the content of this section gives the reader only some basic definitions and paves the way for the following. More details can be found in the references.

Point 3: The results are not significant: it is not clear why k =4,5,6 were chosen even if the elbow rule suggests that the optimal k is k=2. The elbow rule for the SKM is not reported and it does not make sense to show the comparison for k =4,5,6 for KL and for Wasserstein with classical K-means. This way the results are not meaningful.

Response 3: We thank you for pointing out this issue. The cluster number k proposed by the elbow rule is indeed not credible enough on this problem, hence we use an alternative method, i.e., the Gap Statistic. As for the cluster number on the SKM algorithm, we deeply understand your concern. Since there is a one-to-one correspondence between the point clouds on Euclidean space and on manifolds, we choose the same cluster number as that on Euclidean space for the concern of consistence. The descriptions on this issue have also been enriched in the paper.

Point 4: Again about the results, the images make no sense unless it is explained that those are projections in 2 dimensions of a dataset of 6 dimensions (K-means) and 31 dimensions (SKM).

Response 4: We apologize for the negligence of giving explainations about the result images. We actually use PCA to project the high-dimensional points into 2 dimensions. The explaination on this issue has been added to the paper.

Point 5: To validate the procedure one should consider using a known dataset the literature (perhaps using a known repository, e.g. UCI ML) and then use the dataset suggested by the authors.

Response 5: We are grateful for your constructive suggestion. As you suggested, we test the algorithms used in our paper on a UCI ML dataset which has similar data dimensions, and obtain convincing results that support our conclusions. The detailed descriptions about the dataset and the simulation have been added to the end of subsection 5.3.

Reviewer 3 Report

 This paper presents an evaluation of academic performance by using statistical K-means (SKM) algorithm. The main idea is based on the mapping of data from Euclidean space to Riemannian space where the geometric structure of the data is used to obtain accurate clustering results.

In my point of view, the following concerns should be addressed:

·         In the Introduction section, the authors briefly described the problem statement however, they didn’t explain their contribution which is quite important.

·         The paper lacks with the novelty. The idea of transferring the data points from Euclidean space to Riemannian space is used from last two decades (see [1,2]. The authors must specify their contributions.

1.       Faraki, Masoud, Mehrtash T. Harandi, and Fatih Porikli. "More about VLAD: A leap from Euclidean to Riemannian manifolds." Proceedings of the IEEE Conference on Computer Vision and Pattern Recognition. 2015.

2.       Kastaniotis, Dimitris, et al. "Gait based recognition via fusing information from Euclidean and Riemannian manifolds." Pattern Recognition Letters 84 (2016): 245-251.

·         In Section 2, the authors unnecessarily explained a few well-known techniques (e.g., Normal distribution, Wasserstein distance, and etc.) which is not required.

·         How the value of k in k-nearest neighbor is selected?

·         I would suggest the authors to perform the clustering with Gaussian Mixture Models and compare their results with the existing one (see [3]).

3.       Khan, Muhammad Hassan, Muhammad Shahid Farid, and Marcin Grzegorzek. "A generic codebook based approach for gait recognition." Multimedia Tools and Applications 78.24 (2019): 35689-35712.

·         The experimental evaluation in the proposed method is very limited. The experiments needs to be enhanced, especially the number of performed experiments.

·         The computed results must compare with state-of-the-art techniques.

Author Response

Response to Reviewer 3 Comments

Dear Reviewer,

We deeply appreciate the time and efforts you’ve spent in reviewing our manuscript named as “Application of Statistical K-Means Algorithm for University Academic Evaluation” with ID: entropy-1791981. According to your suggestions, we have modified our articles and the revises are as follows.

Point 1: In the Introduction section, the authors briefly described the problem statement however, they didn’t explain their contribution which is quite important.

Response 1: Thanks for your attention, we have added the relative contents in the Introduction section.

Point 2: The paper lacks with the novelty. The idea of transferring the data points from Euclidean space to Riemannian space is used from last two decades (see [1,2]. The authors must specify their contributions.

  1. Faraki, Masoud, Mehrtash T. Harandi, and Fatih Porikli. "More about VLAD: A leap from Euclidean to Riemannian manifolds." Proceedings of the IEEE Conference on Computer Vision and Pattern Recognition. 2015.
  2. Kastaniotis, Dimitris, et al. "Gait based recognition via fusing information from Euclidean and Riemannian manifolds." Pattern Recognition Letters 84 (2016): 245-251.

Response 2: We totally understand your concern and we are very thankful for your reference papers. We have mentioned these related work and their contributions in the introduction. We’d like to clarify that, though the idea of transferring data points from Euclidean space to Riemannian space is not new, there is few previous work about using SPD(n) to extract features and using non-Euclidean metrics as difference function for clustering. Also, not very many statistical methods were used in university academic evaluation, and our paper propose to use statistical tools and clustering algorithm to reveal the underlying relationships of universities, which provides a new research idea for this field.

Point 3: In Section 2, the authors unnecessarily explained a few well-known techniques (e.g., Normal distribution, Wasserstein distance, and etc.) which is not required.

Response 3: Your suggestion is appreciated. We have adjusted the content of Section 2, retaining the parts that are essential to the understanding of this paper for readers unfamiliar with information geometry. We would like to explain that since our algorithm operates on manifolds rather than Euclidean spaces, we introduce the normal distribution manifold (a type of statistical manifold). This will give readers unfamiliar with information geometry a general understanding of manifolds. And the explanation of Wasserstein distance and etc. is to elicit our treatment of its geometric mean.

Point 4: How the value of k in k-nearest neighbor is selected?

Response 4: We apologize for the negligence of explaining the selection of k in k-nearest neighbor. The k is a hyper parameter and there is no specific rule for selecting an exact k. In our paper we select k=10 because every university have 10 data points, and k=10 makes it possible that points from the same university are included in the same sub-cloud, which consists with the theory of the SKM algorithm. We have added the explaination in the paper.

Point 5: I would suggest the authors to perform the clustering with Gaussian Mixture Models and compare their results with the existing one (see [3]).

  1. Khan, Muhammad Hassan, Muhammad Shahid Farid, and Marcin Grzegorzek. "A generic codebook based approach for gait recognition." Multimedia Tools and Applications 78.24 (2019): 35689-35712.

Response 5: Sincere thanks to your suggestion. We have added the Gaussian Mixture Model part in our paper and make comparisons. It proves to be a good reference model.

Point 6: The experimental evaluation in the proposed method is very limited. The experiments needs to be enhanced, especially the number of performed experiments.

Response 6: We agree with your concern and experiments in the paper have been enhanced. We increase the times of each simulation from 5 to 30, and make limits to the maximal iteration times to avoid bad initialization. Corresponding images and tables have also been updated in the paper.

Point 7: The computed results must compare with state-of-the-art techniques.

Response 7: It is true that you suggest our methods should be validated by comparison with other state-of-the-art techniques. We have added an extra simulation on a UCI ML dataset and compared the results with the method proposed by the dataset provider. The results show our advantage. The simulation and descriptions are at the end of the subsection 5.3.

Round 2

Reviewer 2 Report

Dear authors,

The choice of using the Gap statistic is certainly appreciable. however, in order to determine the real number of clusters one should take into account the K that maximizes the Gap diff, as you reported in line 270. According to Figure 5-3, we have two potential values: K=4 or K=6. There are several criteria for choosing 4 or 6 but, given the type of the paper, we can analyze both.

CERTAINLY, K = 5 is not correct. 

The proposed analysis remained the same as the previous time and should be correct.

Good results of the SKM algorithm on the datasets taken from the UCI repository; however, why did you use accuracy for those dataset instead of using DBI, DI and SC like in the Chinese University dataset?  In order to fully compare the results, you should report the values of those indexes, too.

Author Response

Dear Reviewer,

    Sincere appreciation for your admitting of our work and thanks again for spending so much time and efforts on our manuscript named as “Application of Statistical K-Means Algorithm for University Academic Evaluation”. We have carefully read your latest comments and made point-by-point revises and responses. The points are as follows.

Point 1: The choice of using the Gap statistic is certainly appreciable. however, in order to determine the real number of clusters one should take into account the K that maximizes the Gap diff, as you reported in line 270. According to Figure 5-3, we have two potential values: K=4 or K=6. There are several criteria for choosing 4 or 6 but, given the type of the paper, we can analyze both.

CERTAINLY, K = 5 is not correct.

The proposed analysis remained the same as the previous time and should be correct.

Response 1: We are very sorry for taking your extra time on the same issue, and we should apologize for the unclear description and improper presentation on how K is selected. As we explain in the Line 251-261, we choose K=4,5,6 by using Gap Statistic, or Gap Diff exactly. But only one case in which the diffs of K=4 and K=6 are bigger than 0 is given, because it is the most frequent case. We also choose K=5 as this case also proves to appear frequently enough, and more importantly, by using a continuous sequence of K we can observe the variation of clustered universities and measure the relationships among them. It is also more convenient to analyze the trend of cluster assessment indicators with a continuous K.

To make up, we conduct another 50 Gap Statistic simulations and present the results as figure 5-3. The results prove it reasonable to use 5 as cluster number, especially under the considerations mentioned above.

Point 2: Good results of the SKM algorithm on the datasets taken from the UCI repository; however, why did you use accuracy for those dataset instead of using DBI, DI and SC like in the Chinese University dataset?  In order to fully compare the results, you should report the values of those indexes, too.

Response 2: Your suggestion is quite insightful. We only compared the accuracy score because only accuracy scores were given in the paper by the dataset provider. As you sugguested, cluster assessment indicators should be taken into account to better compare the results. We have added a table containing DBI, DI and SC. Different algorithms have basically similar scores on the Chinese University dataset. Euclidean-based methods have better DBI scores while non-Euclidean methods have advantage over SC scores.

Reviewer 3 Report

The most of the concerns are appropriately answered by the authors.

Author Response

Dear Reviewer,

Sincere appreciation for your admitting of our work. Your suggestions are all valuable and have inspiring significance to our researches. Thanks again for spending so much time and efforts on our manuscript named as “Application of Statistical K-Means Algorithm for University Academic Evaluation”.

Point 1: The most of the concerns are appropriately answered by the authors.

Response 1: We are deeply grateful for the comments.

Round 3

Reviewer 2 Report

  1. English typos:
  2.  
  3. The manifold of a family of multivariate normal

  4. 71  distributions is an important statistical manifold and is widely applied to the researches of

  5. 72  signal proseccing -> processing, image proseccing -> processing, neural networks and so on. The K-means algorithm

  6. 73  on statistical manifolds introduced in this paper is to transform the data point cloud in

  7. ........

Check the whole manuscript for other English typos